# Seroprevalence of Measles Antibodies in a Highly MMR-Vaccinated Population

**DOI:** 10.3390/vaccines10111859

**Published:** 2022-11-03

**Authors:** Huy Quang Quach, Inna G. Ovsyannikova, Diane E. Grill, Nathaniel D. Warner, Gregory A. Poland, Richard B. Kennedy

**Affiliations:** 1Mayo Clinic Vaccine Research Group, Division of General of Internal Medicine, Mayo Clinic, Rochester, MN 55905, USA; 2Department of Quantitative Health Sciences, Mayo Clinic, Rochester, MN 55905, USA

**Keywords:** seroprevalence, serosurveillance, measles virus, MMR vaccine, waning immunity

## Abstract

As an extremely contagious pathogen, a high rate of vaccine coverage and the durability of vaccine-induced immunity are key factors to control and eliminate measles. Herein, we assessed the seroprevalence of antibodies specific to measles in a cohort of 1393 adults (20–44 years old). ELISA results showed a nontrivial proportion of 37.6% study subjects being negative for measles immunoglobulin G (IgG). We also found significant influences of sex and age of the study cohort on the IgG level. Our findings suggest that even within a highly vaccinated population, a subset of individuals may still have sub-optimal immunity against measles and potentially be susceptible during any future measles outbreaks.

## 1. Introduction

As one of the most contagious known human diseases, measles infection was an experience affecting essentially everyone during early childhood in the pre-vaccine era, resulting in substantial rates of child mortality and morbidity [1]. However, after the introduction of live, attenuated measles vaccines in 1963 and measles–mumps–rubella (MMR) vaccines in 1971, measles incidence declined by more than 99% [1], leading to the declaration of measles elimination from the United States in 2000 [2]. However, measles has re-emerged with a total of 22 measles outbreaks and 1249 cases reported in 2019 in the United States [3]. Globally, measles outbreaks continue to occur in countries where the population is unvaccinated or under-vaccinated, as well as in countries with highly vaccinated populations [4]. Many countries have suspended their measles campaigns due to the COVID-19 pandemic, which will inevitably lead to an increase in measles cases within those countries [5,6]. Global travel also increases the risk of widespread measles outbreaks.

It is well-documented that MMR vaccine is 93% and 97% effective against measles after one and two doses of vaccination, respectively. Studies have suggested that immunity against measles is persistent, but in a context where subclinical boosting from wild-type virus still exists [7,8]. However, a recent meta-analysis has found that vaccine induced measles antibodies waned at an overall rate of 0.009 annually [9]. Interestingly, measles antibodies in subjects who received two doses of measles vaccine waned in a time-dependent manner with annual rates of 121.8 mIU/mL within 5 years post-vaccination, to 24.1 mIU/mL at 15–20 years post-vaccination [10]. Since vaccine-induced immunity is the sole protector from measles, seroprevalence against measles and its waning needs to be carefully monitored.

In this study, we assessed the seroprevalence of measles-specific antibodies in a cohort of 1393 adults (aged 20–44 years) in Olmsted County, Minnesota (USA), a geographic area where vaccination rates have been extremely high (>95%) and cases have been low-to-nonexistent for decades. We also aimed to identify possible demographic factors associated with the seroprevalence of measles antibodies. Our data showed that a nontrivial proportion of the study cohort had negative measles antibodies and that demographic variables had significant influences on measles antibody titers.

## 2. Methods

### 2.1. Biobank Serum Samples

For this observational study, 2000 subjects who deposited >0.5 mL serum in the Mayo Clinic Biobank were invited to participate in this study. Of the 2000 subjects contacted, 1393 subjects agreed to have their serum samples used for this study. Serum samples (n = 1393) were obtained from the Mayo Clinic Biobank (Rochester, MN, USA), as detailed in a previous report [11]. All subjects were ≥18 years of age and enrolled in the Biobank without any restrictions on health status applied. Height and weight measurements were available in the Biobank for the calculation of body mass index (BMI) of 939 subjects. Based on BMI, study subjects were defined as: underweight (BMI <18.5, n = 16), healthy weight (18.5 to <25, n = 412), overweight (25 to <30, n = 252), and obese (≥30, n = 259).

### 2.2. Measurement of Measles-Specific IgG

Measles-specific IgG antibodies were quantified in serum samples using the Zeus ELISA Measles IgG Test System (Zeus Scientific, Inc., Branchburg, NJ, USA), following the instruction provided. In each ELISA kit, a calibrator and a correction factor were also included. Measles-specific IgG was calculated on the basis of optical density (*OD*) and expressed as a sample index using a provided formula:$$sample\;index=\frac{OD\;of\;testing\;sample}{Mean\;OD\;of\;calibrator\times correction\;factor\;}$$

Based on the sample index, serum samples were classified as positive (sample index ≥1.1), negative (≤0.9) and equivocal (>0.9 to <1.1). According to the manufacturer, the sensitivity and specificity of the assay are 93.3% and 97.4%, respectively. Meanwhile, the coefficient of variability (CV) is 6.7% and 7.2% for intra-assay and interassay, respectively.

The titers of measles-specific IgG were calculated using the Third WHO International Standard Serum for Anti-Measles serum (NIBSC code: 97/648) as standard and expressed in mIU/mL. Based on criteria for disease susceptibility [12], subject’s samples were classified as measles-susceptible (≤120 mIU/mL) or not susceptible to measles (>120 mIU/mL).

### 2.3. Statistical Analysis

Results of continuous variables are presented as median (25th percentile, 75th percentile). Wilcoxon ranked sum test was used to determine a significant difference in measles-specific IgG between males and females while the difference in the measles-specific IgG among the four BMI groups was compared by Kruskal–Wallis test. The association between age and measles-specific IgG was assessed by Spearman’s method. All analyses and figures were conducted in RStudio (version 2022.02.3).

## 3. Results

### 3.1. Demographics of Study Cohort

The demographic characteristics of the study cohort were reported in our previous report [11]. Briefly, the cohort consisted of 1393 subjects with a median age of 36.8 (32.6, 40.8), ranging from 20 to 44 years at the time of serum collection. The cohort had more females (n = 1117; 80.2%) than males (n = 276; 19.8%). Reflecting the demographic population of Olmsted County, MN, USA, 95.3% (n = 1327) of the cohort were Caucasian and 97.6% (n = 1360) were not Hispanic or Latino. The members of the cohort (n = 939) with available BMI data had a median BMI of 25.5 (22.5, 30.9), widely ranging from 13.7 to 58.8. Vaccination records existed for 246 (17.7%) individuals, of which 216 and 30 subjects received one and two doses of MMR vaccine, respectively. Unfortunately, the status of MMR vaccination was unknown for the majority (n = 1147; 82.3%) of the study cohort.

### 3.2. Seroprevalence of Measles-Specific IgG

Measles-specific IgG was measured in sampled serum from each subject by ELISA, and the IgG levels were expressed as the sample index (Figure 1A). The cohort had a median sample index of 1.17 (0.65, 1.96), which was marginally above the positive threshold of 1.1. Based on sample index, 744 (53.4%) serum samples were identified as seropositive (sample index ≥ 1.1) while a significant portion of serum samples was negative (≤0.9; n = 524; 37.6%) or equivocal (between 0.9 and 1.1; n = 125; 9%) (Figure 1A).

The median sample indices of 1.15 (0.69, 2.01), 0.93 (0.64, 1.93), and 1.18 (0.65, 1.94) were calculated for the one-dose, two-dose, and unknown-dose subcohorts, respectively (Figure 1B). The differences in the sample index among the three subcohorts were not significant (*p* = 0.93, Figure 1B). In addition, the one-dose and unknown-dose subcohorts had similar proportions of sample index distribution (Figure 1C). As such, in the one-dose subcohort (n = 216), 83 (38.4%), 20 (9.3%), and 113 (52.3%) serum samples were classified as negative, equivocal, and positive, respectively. Meanwhile, the unknown-dose subcohort (n = 1147) had 426 (37.1%), 104 (9.1%), and 617 (53.8%) negative, equivocal, and positive serum samples, respectively (Figure 1C). The two-dose subcohort (n = 30) had more negative serum samples than positive ones with 15 (50%) and 14 (46.7%) serum samples being classified negative and positive, respectively, while only one serum sample (3.3%) was classified as equivocal.

Using the standard anti-measles serum as a control, a median titer of 158 (76.6, 280) mIU/mL was calculated for the cohort. Using a titer of 120 mIU/mL as the threshold of disease susceptibility [12], 542 (38.9%) and 851 (61.1%) serum samples were classified as susceptible and not susceptible, respectively (Appendix A). The median titers of 155 (82.2, 288), 120 (75.4, 275), and 159 (76.5, 278) mIU/mL were calculated for the one-dose, two-dose, and unknown-dose subcohorts, respectively, and were not significantly different (*p* = 0.84, Appendix A).

### 3.3. Association of Measles-Specific IgG with Age, Sex, and BMI

We assessed associations between demographic factors (sex, age, and BMI) and measles-specific IgG expressed as sample index (Figure 2). We found that the sample index was significantly higher in females than males with a median of 1.2 (0.68, 2.02) and 1.08 (0.59, 1.82) in female and male groups, respectively (*p* = 0.013, Figure 2A). The sample index significantly declined with age of the cohort (R = −0.12, *p* < 0.001, Figure 2B). The sample index was not correlated with BMI (data not shown). Although the overweight group had a lower median sample index of 0.97 (0.55, 1.9) than the underweight, healthy, and obese groups, with median sample indices of 1.22 (0.93, 2.06), 1.23 (0.69, 1.97), 1.18 (0.68, 2.05), respectively, the difference in the sample index among these groups did not reach statistical significance (*p* = 0.097, Figure 2C).

## 4. Discussion

Although measles outbreaks mainly occur in unvaccinated or under-vaccinated populations around the world, vaccine failure (primary failure) and waning immunity (secondary failure) may be contributing factors to measles outbreaks in immunized populations [13,14]. In this serosurveillance study, we assessed the seroprevalence of measles-specific antibodies in 1393 subjects in Olmsted County, MN, USA, and further explored potential influences of demographic factors, including sex, age, and BMI on the measles-specific IgG level. Our results showed that a nontrivial proportion of the study cohort had negative measles IgG, and that sex and age of the study cohort significantly influenced the levels of measles-specific IgG.

A non-significant difference in the levels of measles-specific IgG among three vaccination status subcohorts (i.e., one-dose, two-dose, and unknown-dose) suggested that the majority of this unknown-dose subcohort had been vaccinated with at least one dose of MMR vaccine (Figure 1C). In fact, the distribution of the sample index was similar between one-dose and unknown-dose subcohorts, further supporting the hypothesis that the majority of the unknown-dose subcohort had received at least one dose of MMR vaccine. Furthermore, two doses of MMR vaccine were recommended for those aged 4–6 years in the United States in 1989 [15]. With a median age of 36.8 years at the time of serum collection, the individuals of unknown-dose subcohort likely received at least one dose of MMR vaccine. Interestingly, the proportion of negative serum samples in the two-dose subcohort was higher than those in the one-dose and unknown-dose subcohorts (Figure 1C). This could be explained by a high rate of measles antibody decay following a second dose of MMR vaccine, as observed earlier [16]. Along these same lines, it is possible that this cohort had more complete records, and did not contain individuals missing a more recent vaccination. The small size of the two-dose subcohort (n = 30) may also be a factor as sampling bias may exist, and this small cohort is not truly reflective of all two-dose MMR recipients in the community. 

Even in those with known vaccination history, 38.4% (in the one-dose subcohort) or 50% (in the two-dose subcohort) of individuals had negative measles IgG. This could be explained by waning measles antibodies, or by either improper vaccination technique or outdated/expired vaccine, both of which are unlikely. The former hypothesis is supported by the negative association between age and measle-specific IgG level observed in this study (Figure 2B) and is consistent with other published reports [9,10,13,16,17]. Note that 97.8% of subjects in the same study cohort were found seropositive for rubella-specific IgG [11]. Since rubella is one of viral components of the MMR vaccine, the high percentage (97.8%) of rubella-specific, seropositive subjects in this cohort suggests that the majority of study cohort have received at least one dose of MMR vaccine, further supporting our hypothesis of waning measles immunity.

Sex is a well-known biological variable that influences vaccine-induced immune responses [18]. Consistent with previous report [19], females in this study cohort had higher measles-specific IgG level than males (Figure 2A). Although this observation could be confounded by a higher ratio of females (80.2%) than males (19.8%) in the study cohort of this study, the large sample size (n = 1393) could be sufficient for a valid comparison [20]. With that regard, the higher measles-specific IgG level in females could be explained by a stronger initial humoral immune response to measles vaccines [21] and/or a slower waning of measles immunity in females than in males [19]. 

Aging heavily affects the activity of the immune system, hence age is an important demographic factor influencing the immune responses induced by infection or vaccination [22]. Ideally, the time since the last MMR vaccination to the time of collecting serum samples should be examined. Unfortunately, the record of vaccination history was not available in the Biobank for the majority of the study cohort. Assuming the study subjects followed the timeframe of MMR vaccination recommended in the United States, the age of the study cohort at the time of serum collection could be examined as a surrogate of the time post-vaccination. A negative correlation between measles-specific IgG level and age of the study cohort at the time of serum collection (R = −0.12, *p* < 0.001) implies the waning of measles immunity as previously reported [9,10,13,16,17]. 

This serosurveillance study has several limitations; therefore, the findings should be cautiously interpreted. First, the subjects included in this survey were regional (Olmsted County, MN, USA) rather than representative of the United States population. Indeed, 80.2% of study cohort were females, almost all, 95.3% and 97.6%, were Caucasian and not Hispanic or Latino, respectively. A diverse cohort may provide a better general insight applicable to the entire United States population. Second, cellular responses are known to play a role in the clearance of viral RNA [23], but we did not evaluate measles-specific cellular immune responses in this study. Together with measles-specific antibody, cellular responses will provide a comprehensive picture of measles immunity. Note that while the absence of circulating measles virus supports our working assumption that measured antibody values are vaccine-induced, it also makes it difficult to conclude that the population lacks immunity to measles. Third, information on the health status of the study subjects was limited. Comorbidity is a potential confounding factor that influences the immune outcomes observed in this survey; hence, future studies should take this factor into consideration.

Meanwhile, the results from this study have important implications for MMR immunization strategy for the adult population. First, since measles-specific antibodies wane faster in males than females, the timeframe for a boosting dose of MMR vaccine should be shorter for males than females. Second, under recent mumps outbreaks, a boosting dose of MMR vaccine is recommended for population with a high rate of MMR coverage [24]. However, data from our previous study showed that the boosting dose of MMR vaccine only enhanced measles-specific immune responses in individuals with negative or low measles-specific antibodies; individuals with a positive or high titer of measles-specific antibodies did not benefit from the boosting dose of MMR vaccine [25]. Therefore, additional factors need to be taken into consideration when the boosting dose of MMR vaccine is considered. For example, an additional screening step of measles-specific antibodies should be taken before immunization and the boosting dose should practically be given to individuals with negative or low measles-specific antibodies. However, this screening step may be time-consuming and costly. Under certain circumstances such as outbreaks, it may faster to achieve herd immunity by boosting all individuals at risk.

In summary, we assessed the measles seroprevalence in 1393 young adults aged 20–44 years in this study. We observed a nontrivial proportion of study subjects having negative measles-specific IgG. We also found significant influences of sex and age of study subjects on the titer of measles-specific IgG. Our findings highlight a suboptimal level of measles immunity in the studied cohort, suggesting a potential high risk of measles outbreaks. Our findings also suggest a possibility of revisiting the program of MMR vaccination to increase herd immunity.

## Figures and Tables

**Figure 1 vaccines-10-01859-f001:**
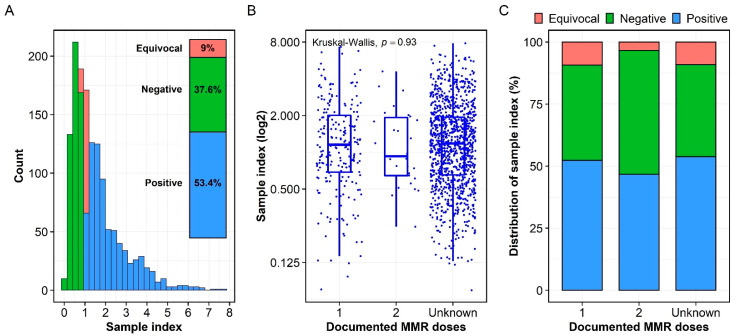
Measles-specific IgG antibodies. (**A**) Distribution of measles-specific IgG antibodies expressed as the sample index. On the basis of the sample index, 744 (53.4%), 524 (37.6%), 125 (9%) serum samples were identified as positive, negative, and equivocal, respectively (Figure 1A inset). (**B**) There was no significant difference in the IgG levels as expressed as sample index among subjects with one-dose, two-dose and unknown-dose MMR vaccination. (**C**) Distribution of the sample index in the one-dose, two-dose, and unknown-dose subcohorts. While the one-dose and unknown-dose subcohorts had similar distribution of the sample index, the two-dose subcohort had more negative serum samples than positive ones.

**Figure 2 vaccines-10-01859-f002:**
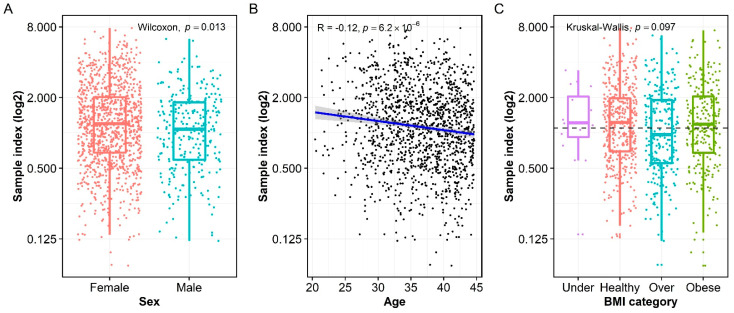
The influences of demographic variables on measles-specific IgG titer. (**A**) Females had a significantly higher median sample index than males. (**B**) Measles-specific IgG level was negatively correlated to the age of the cohort. (**C**) Overweight individuals (labelled as “Over”) had significantly lower IgG titer than underweight (labelled as “Under”), healthy, and obese individuals, but the difference did not reach statistical significance.

## Data Availability

Raw data and R codes used to analyze the data are deposited to Synapse (http://www.synapse.org), with Synapse ID of syn42828599 (https://www.synapse.org/#!Synapse:syn42828599/wiki/619746), accessed on 31 October 2022.

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
