# Peer review of "Seroprevalence of Measles Antibodies in a Highly MMR-Vaccinated Population"

_vaccines, 2022, doi:10.3390/vaccines10111859_

Round 1

Reviewer 1 Report

Estimated Dr. QUACH, 

I've read with great interest this short but very well written and informative paper on the failure in measles immunization from the Olmsted county in the US. Briefly, the cohort consisted of 1,393 subjects with a median age of 36.8, and a "non trivial" share of the sampled individuals (i.e. 37.6% + 9%) had serum samples that suggested they potential vulnerability to the virus.

Authors have tentatively analyzed their sample and, based on available data, suggested some explanations (i.e. age, gender, BMI) for these vaccination failure. Even though not-resolutive, this study may represent a well designed starting point for future assessment of the risk factors for MMR vaccine failures. Therefore, I not only recommend but also urge for the acceptance of this study in its current form.

Author Response

Dear Reviewer,

Thank you very much for your time reviewing our manuscript. Since you did not have any specific comments/request for the revision, we uploaded the revised manuscript for your reference. Please do not hesitate to contact me if any questions arise.

Sincerely,

Richard B. Kennedy, Ph.D.

The point-to-point response to reviewer’s comments:

Reviewer 1:
Estimated Dr. QUACH,
1. I've read with great interest this short but very well written and informative paper on the failure in measles immunization from the Olmsted county in the US. Briefly, the cohort consisted of 1,393 subjects with a median age of 36.8, and a "non trivial" share of the sampled individuals (i.e. 37.6% + 9%) had serum samples that suggested they potential vulnerability to the virus.
Authors have tentatively analyzed their sample and, based on available data, suggested some explanations (i.e. age, gender, BMI) for these vaccination failure. Even though not-resolutive, this study may represent a well designed starting point for future assessment of the risk factors for MMR vaccine failures. Therefore, I not only recommend but also urge for the acceptance of this study in its current form.

Author’s response: Thank you very much for your time reviewing this manuscript.

Reviewer 2 Report

I read the manuscript entitled "Seroprevalence of measles antibodies in a highly MMR-vaccinated population". The idea is not very new! But the overall finding is a bit interesting. Usually, the sensitivity and specificity of the serological tests were low. How were the sensitivity and specificity of your test? Please clarify those in your methods. How do you differentiate antibodies from natural infections and vaccinations? Justify your results in terms of natural infection and/or vaccination-acquired antibodies. The overall writing is lack integrity and coherence and needs improvement. See some specific comments below: 

Abstract: I see the abstract and the end part of the discussion as the same/duplicate. I suggested deleting the discussion's end portion and replacing it with conclusions pointing out your key findings.

Methods: Add ethical approval number as this was missing. Add a separate bland consent form as a supplementary file.

Add your data set and R-code/command as a supplementary file.

Author Response

Dear Reviewer,

Thank you very much for your time reviewing our manuscript and your constructive comments and suggestions. We strongly believe that your comments and suggestions greatly strengthen this manuscript. We have considered your comments/suggestions in depth and responded to each of your comments/suggestions in detail as below. We have also revised the manuscript accordingly and uploaded the revised manuscript for your reference. All changes in the revised manuscript were tracked. We have also uploaded our responses to the comments/suggestions from other reviewers for your information. Please do not hesitate to contact me if any questions arise.

Sincerely,

Richard B. Kennedy, Ph.D.

The point-to-point response to reviewer’s comments:
Reviewer 2:
I read the manuscript entitled "Seroprevalence of measles antibodies in a highly MMR-vaccinated population". The idea is not very new! But the overall finding is a bit interesting. Usually, the sensitivity and specificity of the serological tests were low. How were the sensitivity and specificity of your test? Please clarify those in your methods. How do you differentiate antibodies from natural infections and vaccinations? Justify your results in terms of natural infection and/or vaccination-acquired antibodies. The overall writing is lack integrity and coherence and needs improvement. See some specific comments below.

Author’s response: Thank you the reviewer for the comment. We agree with the reviewer that the idea in our manuscript is not very new. The waning of measles immunity has been reported in several publications [1,2], as mentioned in the Introduction of this manuscript. In this study, we measured the measles-specific IgG in a large study cohort of 1,393 subjects. Briefly, we found that a significant (37.6%) proportion of study subjects had negative measles-specific IgG. In addition to confirming the waning of measles-specific IgG in this large cohort, we found that i) female had higher measles-specific IgG than male; ii) there was no significant difference in the level of measles-specific IgG between one-dose and two-doses subcohorts. We believe that these informative results are useful to develop a better strategy for measles vaccination program.

As described in section 2.3 (line #67 - #83 in the revised manuscript), we used a commercially available ELISA kit to detect measles-specific IgG. According to the manufacturer, the sensitivity and specificity of the assay are 93.3% and 97.4%, respectively, with the intra-assay and interassay coefficient of variability (CV) of 6.7% and 7.2%, respectively. We have revised section 2.3 to include the sensitivity and specificity of the assay, as suggested. Please kindly check line #77 - #78 in the revised manuscript.

In this study, we did not differentiate the measles-specific IgG induced by vaccination from that by natural infection. However, medical records of study subjects did not show any cases of measles infection in our study cohort. In addition, Measles Disease Statistics reported by Minnesota Department of Health show that there is a total of 142 measles infection case during the 2000 -
2022 period in Minnesota and none of these cases occurs in Olmsted County [3]. The vast majority of the cohort are local, long-time residents of Olmsted County and we believe that the measles-specific IgG reported for these individuals is likely induced by vaccination. We cannot completely exclude the possibility that some individuals experienced measles infection in the past, although we believe these cases would be rare as we found documentation of such events in the medical records. We agree with the reviewer that it is important to differentiate and justify IgG results in terms of natural infection and/or vaccination-acquired antibodies, but it is unlikely the case in this study.

We have gone through the entire manuscript in order to clarify language as necessary to make the text more integral and coherent.

1. Comment to the authors: Abstract: I see the abstract and the end part of the discussion as the same/duplicate. I suggested deleting the discussion's end portion and replacing it with conclusions pointing out your key findings.

Author’s response: Thank you the reviewer for raising this point. However, the end portion of Discussion is the Conclusion of the manuscript (in original submitted version); therefore, it contains key findings of this work, which are also highlighted in the Abstract. Please kindly check the Conclusion (line #244 - #250) and the Abstract (line #12 - #19) in the revised manuscript.

2. Methods: Add ethical approval number as this was missing. Add a separate bland consent form as a supplementary file.

Author’s response: Thank you the reviewer for raising this point. We have added the ethical approval number in the revised manuscript. Please kindly check line #55 - #56 in the revised manuscript.
We have also uploaded a blank consent form for your reference. However, we decide to not publish this form as a part of Supplementary Materials because Mayo Clinic considers this form an internal work product and does not publish consent forms for non-clinical trial studies. We seek your understanding on this point. Please kindly find the consent form uploaded for your reference.

3. Add your data set and R-code/command as a supplementary file.

Author’s response: Thank you the reviewer for the suggestion. The research reported in this manuscript is funded by the National Institutes of Health (NIH). As indicated in the NIH agreement, data generated using NIH funding needs to be uploaded. Therefore, we will upload the dataset used in this manuscript to ImmPort and indicate that the dataset is published in this manuscript, if accepted. In response to this request, we have uploaded the dataset and R code used in this study to Synapse (http://www.synapse.org), with Synapse ID of syn42828599. Note that to protect the anonymity and confidentiality of study participants, sample ID was de-identified in the uploaded dataset. We have also revised the manuscript to include the data availability statement. Please kindly check line #91 - #93 in the revised manuscript.

References

1. Bolotin, S.; Osman, S.; Hughes, S.L.; Ariyarajah, A.; Tricco, A.C.; Khan, S.; Li, L.; Johnson, C.; Friedman, L.; Gul, N.; et al. In Elimination Settings, Measles Antibodies Wane Following Vaccination but Not Following Infection - A Systematic Review and Meta-Analysis. J Infect Dis 2022, doi:10.1093/infdis/jiac039.

2. Schenk, J.; Abrams, S.; Theeten, H.; Van Damme, P.; Beutels, P.; Hens, N. Immunogenicity and persistence of trivalent measles, mumps, and rubella vaccines: a systematic review and meta-analysis. Lancet Infect Dis 2021, 21, 286-295, doi:10.1016/s1473-3099(20)30442-4.

3. Available online: https://www.health.state.mn.us/diseases/measles/stats.html (accessed on

Reviewer 3 Report

This brief report by Huy Quang Quach et al. analyzes the seroprevalence of measles antibodies in a highly vaccinated population in the Unit States. The findings of this report suggest a progressive decline of the measles immune barrier in an MMR vaccinated population. This is consistent with the multiple outbreaks in recent years following the declaration of measles elimination from the USA in 2000. Overall, the findings of this report were well presented, only required minor revisions before publication.

Comments:

1.       It is recommended that the sample index should be convert to international units (mIU/ml), for comparison with other studies.

2.       The important findings in this report were based on the hypothesis that the majority of the unknown-dose cohort have been vaccinated with at least one dose of MMR. Because the measles vaccine history of most enrolled participants in this report is unknown. Therefore, the age, sex distribution between vaccinated and unvaccinated groups should be carefully compared to support the hypothesis.

3. In the discussion, it is better to give recommendations for the MMR immunization strategies for adult population in the United States based on findings. Such as booster immunization among adolescents in USA

Author Response

Dear Reviewer,

Thank you very much for your time reviewing our manuscript and your constructive comments and suggestions. We strongly believe that your comments and suggestions greatly strengthen this manuscript. We have considered your comments/suggestions in depth and responded to each of your comments/suggestions in detail as below. We have also revised the manuscript accordingly and uploaded the revised manuscript for your reference. All changes in the revised manuscript were tracked. We have also uploaded our responses to the comments/suggestions from other reviewers for your information. Please do not hesitate to contact me if any questions arise.

Sincerely,

Richard B. Kennedy, Ph.D.

The point-to-point response to reviewer’s comments:

Reviewer 3:

This brief report by Huy Quang Quach et al. analyzes the seroprevalence of measles antibodies in a highly vaccinated population in the Unit States. The findings of this report suggest a progressive decline of the measles immune barrier in an MMR vaccinated population. This is consistent with the multiple outbreaks in recent years following the declaration of measles elimination from the USA in 2000. Overall, the findings of this report were well presented, only required minor revisions before publication.

Comments:
1. It is recommended that the sample index should be convert to international units (mIU/mL), for comparison with other studies.

Author’s response: Thank you the reviewer for the suggestion. We have converted the sample index into mIU/mL using the Third WHO International Standard Serum for Anti-measles serum (NIBSC code: 97/648) as reference. These new results are summarized in Supplementary Figure S1. It should be noted that although the optical density (OD) values of WHO standard serum were comparable to these of kit calibrator at the same concentrations, the WHO standard serum is calibrated for plaque reduction neutralization assay, but has not officially been calibrated for the use in ELISA assay. We have revised the manuscript to include these results in the revised manuscript. Please kindly check line #80 - #83, #135 - #141 and Supplementary Figure S1 for more information.

2. The important findings in this report were based on the hypothesis that the majority of the unknown-dose cohort have been vaccinated with at least one dose of MMR. Because the measles vaccine history of most enrolled participants in this report is unknown. Therefore, the age, sex distribution between vaccinated and unvaccinated groups should be carefully compared to support the hypothesis.

Author’s response: Thank you the reviewer for raising this concern. We agree with the reviewer that caution needs to be taken when interpreting the results from our study since
MMR vaccination record is missing for the majority of study subjects. However, measles-mumps-rubella (MMR) and measles-mumps-rubella-varicella (MMRV) are measles-containing vaccines are widely used in the US. In our previous report on the same study cohort [1], we found that 97.8% of study subjects had positive rubella-specific IgG. Since rubella is one of viral components of both MMR and MMRV vaccines, this result suggests that the majority of study subjects received at least one dose of MMR or MMRV vaccine.

3. In the discussion, it is better to give recommendations for the MMR immunization strategies for adult population in the United States based on findings. Such as booster immunization among adolescents in USA.

Author’s response: Thank you the reviewer for suggesting this point. We have added our recommendations for the MMR immunization strategies in the United States in the revised manuscript. Briefly, we suggest that i) boosting timeframe should be different for males and females since vaccine-induced measles-specific antibodies wane faster in males than females; ii) measles-specific IgG should be screened in adult population in the United State before boosting dose; iii) in our previous study [2], we found that a third dose of MMR vaccine did not boost both cellular and humoral measles-specific immune responses in individuals with positive or high titer of measles-specific antibodies; therefore, a booster dose should be logically given to individuals with negative or low measles-specific IgG. We also realize that this strategy is practically challenging since screening measles-specific antibodies in a whole population is quite costly and time-consuming step. Under certain circumstances such as outbreaks, international travels, and in selected workplaces (e.g., hospitals, school environments, military bases) we believe that it is safer and faster to give a booster to all individuals potentially at risk. Please kindly check line #229 - #243 in the revised manuscript.

References

1. Crooke, S.N.; Haralambieva, I.H.; Grill, D.E.; Ovsyannikova, I.G.; Kennedy, R.B.; Poland, G.A. Seroprevalence and durability of rubella virus antibodies in a highly immunized population. Vaccine 2019, 37, 3876-3882, doi:10.1016/j.vaccine.2019.05.049.

2. Quach, H.Q.; Chen, J.; Monroe, J.M.; Ratishvili, T.; Warner, N.D.; Grill, D.E.; Haralambieva, I.H.; Ovsyannikova, I.G.; Poland, G.A.; Kennedy, R.B. The influence of sex, BMI, and age on cellular and humoral immune responses against measles after a 3rd dose of MMR vaccine. J Infect Dis 2022, doi:10.1093/infdis/jiac351.

Reviewer 4 Report

1.       Lines 38-39: typo: “wild-type” virus.

2.       Lines 39-40: please clarify the phrase “…annual waning seroconversion rate of 0.009 in vaccine-induced measles antibodies” – does it mean 0.009 percent annual decrease?

3.       Lines 50-52: the authors should not present their study results in the Introduction.

4.       Lines 75-77: please clarify how these correlates of protection compare with recommended values – see Bolotin et al. What Is the Evidence to Support a Correlate of Protection for Measles? A Systematic Review. Journal of Infectious Diseases 2020;221:1576–83 DOI: 10.1093/infdis/jiz380.

5.       Methods and Discussion: the authors should clarify if data were collected on history of measles infection among the study participants.  Clearly, some of the study subjects, particularly those with unknown vaccination status and who lived elsewhere before moving to Olmstead County, may have had measles infection in their youth.

6.       Lines 174-176: please clarify which study generated the 97% rubella IgG +ve result and why this provides evidence of waning immunity.

7.       Lines 189-194: please clarify if the ‘surrogate’ analysis was in fact carried out and how this correlation demonstrates waning immunity.

8.       Lines 201-204: it has long been established that a very high proportion of individuals who develop antibodies following measles infection or vaccination, but whose antibodies become undetectable over time, still mount a rapid response to a new measles challenge.  Much of the literature confirms that what matters in public health terms is not whether individuals show detectable antibodies but whether they are protected when challenged.

The statement “With the well-established role of cellular responses in preventing measles [22], low measles antibody titer may not necessarily equate to lack of measles immunity. Note that while the absence of circulating measles virus supports our working assumption that measured antibody values are vaccine-induced, it also makes it difficult to conclude that the population lacks immunity to measles.” should be included early in the Introduction, and the authors should clarify why they decided to proceed with this study in view of this critical limitation to the interpretation of such analyses.

Author Response

Dear Reviewer,

Thank you very much for your time reviewing our manuscript and your constructive comments and suggestions. We strongly believe that your comments and suggestions greatly strengthen this manuscript. We have considered your comments/suggestions in depth and responded to each of your comments/suggestions in detail as below. We have also revised the manuscript accordingly and uploaded the revised manuscript for your reference. All changes in the revised manuscript were tracked. We have also uploaded our responses to the comments/suggestions from other reviewers for your information. Please do not hesitate to contact me if any questions arise.

Sincerely,

Richard B. Kennedy, Ph.D.

The point-to-point response to reviewer’s comments:
Reviewer 4:

1. Lines 38-39: typo: “wild-type” virus.

Author’s response: Thank you the reviewer for pinpointing this typo. It has been corrected in the revised manuscript. Please kindly check line #38 - #39 in the revised manuscript.

2. Lines 39-40: please clarify the phrase “…annual waning seroconversion rate of 0.009 in vaccine-induced measles antibodies” – does it mean 0.009 percent annual decrease?

Author’s response: Thank you the reviewer for raising this point. Yes, the phrase “annual waning seroconversion rate of 0.009” means that vaccine-induced measles antibodies decrease at an annual rate of 0.009 [1]. We have rephrased to clarify this point in the revised manuscript. Please kindly check line #39 - #40 in the revised manuscript.

3. Lines 50-52: the authors should not present their study results in the Introduction.

Author’s response: Thank you the reviewer for the suggestion. We have removed the study results in the Introduction. Please kindly check line #50 - #51 in the revised manuscript.

4. Lines 75-77: please clarify how these correlates of protection compare with recommended values – see Bolotin et al. What Is the Evidence to Support a Correlate of Protection for Measles? A Systematic Review. Journal of Infectious Diseases 2020;221:1576–83 DOI: 10.1093/infdis/jiz380.

Author’s response: Thank you the reviewer for providing us this reference. A measles IgG titer of >120 mIU/mL (as measured by plaque reduction neutralization test) is assumed to provide protection from measles disease [2]. However, this serologic correlate of protection is being challenged, as reviewed recently [3]. Therefore, further work is needed to validate this correlate of protection. We have included these two studies in our manuscript as references.
In this study, we used a commercial ELISA kit to detect measles IgG. Each ELISA kit has the calibrator and correction factor to calculate the sample index. The kit calibrator has a range of 200 - 350 mIU/mL measles IgG. Therefore, a sample of >1.1 would have a measles IgG titer >120 mIU/mL, assuming that this correlate of protection is still valid. We have included the titer of measles-specific IgG in mIU/mL in the revised manuscript. Please kindly check line #80 - #83, #135 - #141 and Supplementary Figure S1 for more information.

5. Methods and Discussion: the authors should clarify if data were collected on history of measles infection among the study participants. Clearly, some of the study subjects, particularly those with unknown vaccination status and who lived elsewhere before moving to Olmstead County, may have had measles infection in their youth.

Author’s response: Thank you the reviewer for raising this point. Basing on medical history, there is no measles infection case recorded in our study cohort. It is possible that some study subjects had been infected before moving to Olmsted County. However, according to Measles Disease Statistics, there is a total of 142 measles infection case in Minnesota and none of them is reported in Olmsted County during the 2000 - 2022 period [4]. Therefore, we believe that measles infection is not likely the case of our study cohort.

6. Lines 174-176: please clarify which study generated the 97% rubella IgG +ve result and why this provides evidence of waning immunity.

Author’s response: Thank you the reviewer for raising this point. In this study, we used serum samples from the same study cohort, as previously reported by our group [5]. In that report [5], we found that 97.8% of study subjects had positive rubella IgG. Since rubella is a viral component of MMR vaccine, this result suggest that majority of study subjects received at least one dose of MMR vaccine. Therefore, together with negative correlation between age and measles-specific IgG level observed in this manuscript, this rubella-specific IgG results support our hypothesis of waning measles immunity.
We have revised the manuscript to better this point. Please kindly check line #191 - #197 in the revised manuscript.

7. Lines 189-194: please clarify if the ‘surrogate’ analysis was in fact carried out and how this correlation demonstrates waning immunity.

Author’s response: Thank you the reviewer for raising this point. From line #189 - #194, we mean that the age of study cohort can be used as a surrogate of the time of last MMR vaccination with an assumption that study subjects follow the timeframe of MMR vaccination recommended in the US. In that case, a negative correlation between age and measles-specific IgG level suggests the waning of measles immunity. Ideally, the time since the last MMR vaccination to the time of collecting serum samples should be examined. However, vaccination record is not available for majority of study subjects. We have revised the manuscript to clarify this point. Please kindly check line #211 - #214 in the revised manuscript.

8. Lines 201-204: it has long been established that a very high proportion of individuals who develop antibodies following measles infection or vaccination, but whose antibodies become undetectable over time, still mount a rapid response to a new measles challenge. Much of the literature confirms that what matters in public health terms is not whether individuals show detectable antibodies but whether they are protected when challenged.

Author’s response: Thank you the reviewer for the comment. However, to the best of our knowledge we partly do not agree with the reviewer on this point for some reasons. First, vaccine-induced measles antibody persists for nearly 30 years [6,7]. Second, the waning of measles immunity is mainly characterized by the level of measles antibody in vaccinated individuals [1,8]. If the developed measles antibody become undetectable overtime, it is hard to conclude that it wanes [1,8]. We know it wanes because we can measure it. Third, data from recent measles outbreaks showed a low level of measles antibody in infected individuals [9-12]. If measles antibodies are undetectable over time and mount rapidly to a measles challenge in vaccinated individuals, these individuals would not be infected. Fourth, if the measles-specific antibodies are undetectable in vaccinated individuals, it is not practical to have these individuals challenged with measles virus just to confirm if they are protected.

9. The statement “With the well-established role of cellular responses in preventing measles [22], low measles antibody titer may not necessarily equate to lack of measles immunity. Note that while the absence of circulating measles virus supports our working assumption that measured antibody values are vaccine-induced, it also makes it difficult to conclude that the population lacks immunity to measles.” should be included early in the Introduction, and the authors should clarify why they decided to proceed with this study in view of this critical limitation to the interpretation of such analyses.

Author’s response: Thank you the reviewer for the suggestion. In this study, we measured the level of measles-specific IgG in a large cohort of 1,393 subjects and found that 37.6% of study cohort had negative measles-specific IgG. We also found that the level of measles-specific IgG was negatively correlated with the age of study subjects, suggesting a waning measles immunity. We focused on humoral immunity for two reasons: humoral correlates of protection have been established; and sera samples were available for all study subjects while PBMCs were not. We do agree that these results are not sufficient to draw a conclusion that those study subjects who had low measles-specific IgG lack measles immunity and are susceptible to future infection, because cellular responses also play a protective role [13]. Therefore, this is the limitation of this study and has been noted in the revised manuscript. However, the results observed in this study provide useful information for us to design future works. Basing on the level of measles-specific antibody, we select study subjects with low and high measles-specific antibody levels. Then, we compare measles-specific cellular responses in these two groups. Altogether, these results will provide a more comprehensive picture of measles immunity. We have rephrased the text to better explain this limitation in the manuscript. Please kindly check line #220 - #223 in the revised manuscript.

References
1. Schenk, J.; Abrams, S.; Theeten, H.; Van Damme, P.; Beutels, P.; Hens, N. Immunogenicity and persistence of trivalent measles, mumps, and rubella vaccines: a systematic review and meta-analysis. Lancet Infect Dis 2021, 21, 286-295, doi:10.1016/s1473-3099(20)30442-4.

2. Chen, R.T.; Markowitz, L.E.; Albrecht, P.; Stewart, J.A.; Mofenson, L.M.; Preblud, S.R.; Orenstein, W.A. Measles antibody: reevaluation of protective titers. J Infect Dis 1990, 162, 1036-1042, doi:10.1093/infdis/162.5.1036.

3. Bolotin, S.; Hughes, S.L.; Gul, N.; Khan, S.; Rota, P.A.; Severini, A.; Hahné, S.; Tricco, A.; Moss, W.J.; Orenstein, W.; et al. What Is the Evidence to Support a Correlate of Protection for Measles? A Systematic Review. J Infect Dis 2020, 221, 1576-1583, doi:10.1093/infdis/jiz380.

4. Available online: https://www.health.state.mn.us/diseases/measles/stats.html (accessed on

5. Crooke, S.N.; Haralambieva, I.H.; Grill, D.E.; Ovsyannikova, I.G.; Kennedy, R.B.; Poland, G.A. Seroprevalence and durability of rubella virus antibodies in a highly immunized population. Vaccine 2019, 37, 3876-3882, doi:10.1016/j.vaccine.2019.05.049.

6. Dine, M.S.; Hutchins, S.S.; Thomas, A.; Williams, I.; Bellini, W.J.; Redd, S.C. Persistence of vaccine-induced antibody to measles 26–33 years after vaccination. The Journal of infectious diseases 2004, 189, S123-S130.

7. Ramsay, M.; Moffatt, D.; O'connor, M. Measles vaccine: a 27–year follow–up. Epidemiology & Infection 1994, 112, 409-412.

8. Bolotin, S.; Osman, S.; Hughes, S.L.; Ariyarajah, A.; Tricco, A.C.; Khan, S.; Li, L.; Johnson, C.; Friedman, L.; Gul, N.; et al. In Elimination Settings, Measles Antibodies Wane Following Vaccination but Not Following Infection - A Systematic Review and Meta-Analysis. J Infect Dis 2022, doi:10.1093/infdis/jiac039.

9. Cheng, V.C.; Wong, S.-C.; Wong, S.C.; Sridhar, S.; Chen, J.H.; Yip, C.C.; Hung, D.L.; Li, X.; Chuang, V.W.; Tsang, O.T. Measles outbreak from Hong Kong International Airport to the hospital due to secondary vaccine failure in healthcare workers. Infection Control & Hospital Epidemiology 2019, 40, 1407-1415.

10. Hahné, S.J.; Nic Lochlainn, L.M.; van Burgel, N.D.; Kerkhof, J.; Sane, J.; Yap, K.B.; van Binnendijk, R.S. Measles outbreak among previously immunized healthcare workers, the Netherlands, 2014. The Journal of infectious diseases 2016, 214, 1980-1986.

11. Rosen, J.B.; Rota, J.S.; Hickman, C.J.; Sowers, S.B.; Mercader, S.; Rota, P.A.; Bellini, W.J.; Huang, A.J.; Doll, M.K.; Zucker, J.R. Outbreak of measles among persons with prior evidence of immunity, New York City, 2011. Clinical Infectious Diseases 2014, 58, 1205-1210.

12. Song, K.; Lee, J.M.; Lee, E.J.; Lee, B.R.; Choi, J.Y.; Yun, J.; Lee, S.N.; Jang, M.Y.; Kim, H.W.; Kim, H.-S. Control of a nosocomial measles outbreak among previously vaccinated adults in a population with high vaccine coverage: Korea, 2019. European Journal of Clinical Microbiology & Infectious Diseases 2022, 1-12.

13. Lin, W.H.; Pan, C.H.; Adams, R.J.; Laube, B.L.; Griffin, D.E. Vaccine-induced measles virus-specific T cells do not prevent infection or disease but facilitate subsequent clearance of viral RNA. mBio 2014, 5, e01047, doi:10.1128/mBio.01047-14.